# Bacitracin and Rutin Regulate Tissue Factor Production in Inflammatory Monocytes and Acute Myeloid Leukemia Blasts

**DOI:** 10.3390/cancers13163941

**Published:** 2021-08-04

**Authors:** Lennart Beckmann, Christina Charlotte Rolling, Minna Voigtländer, Jonathan Mäder, Felix Klingler, Anita Schulenkorf, Carina Lehr, Carsten Bokemeyer, Wolfram Ruf, Florian Langer

**Affiliations:** 1II. Medizinische Klinik und Poliklinik, Universitätsklinikum Eppendorf, 20246 Hamburg, Germany; l.beckmann@uke.de (L.B.); c.rolling@uke.de (C.C.R.); m.voigtlaender@uke.de (M.V.); 6958528@stud.uke.uni-hamburg.de (J.M.); f.klingler@uke.de (F.K.); a.schulenkorf@uke.de (A.S.); c.lehr@uke.de (C.L.); cbokemeyer@uke.de (C.B.); 2Department of Medicine, New York University Grossman School of Medicine, New York, NY 10016, USA; 3Centrum für Thrombose und Hämostase, Universitätsmedizin der Johannes Gutenberg-Universität Mainz, 55131 Mainz, Germany; ruf@uni-mainz.de; 4Department of Immunology and Microbiology, The Scripps Research Institute, La Jolla, CA 92037, USA

**Keywords:** tissue factor, protein disulfide isomerase, acute myeloid leukemia, coagulation, inflammation, rutin, monocytes

## Abstract

**Simple Summary:**

Aberrant tissue factor (TF) expression by transformed myeloblasts and inflammatory monocytes contributes to coagulation activation in acute myeloid leukemia (AML). TF procoagulant activity (PCA) is regulated by protein disulfide isomerase (PDI), an oxidoreductase with chaperone activity, but its specific role in AML-associated TF biology is unclear. Here, we provide novel mechanistic insights into this interrelation. We show that bacitracin and rutin, two pan-inhibitors of the PDI family, prevent lipopolysaccharide (LPS)-induced monocyte TF production under inflammatory conditions and constitutive TF expression by THP1 cells and AML blasts, thus exerting promising anticoagulant activity. Downregulation of the TF protein was mainly restricted to its non-coagulant, cryptic pool and was at least partially regulated on the mRNA level in LPS-stimulated monocytes. Collectively, our study indicates a complex role of thiol isomerases in the regulation of myeloid TF PCA, with the most abundant PDI being a promising therapeutic target in the management of AML-associated coagulopathies.

**Abstract:**

Aberrant expression of tissue factor (TF) by transformed myeloblasts and inflammatory monocytes drives coagulation activation in acute myeloid leukemia (AML). Although regulation of TF procoagulant activity (PCA) involves thiol-disulfide exchange reactions, the specific role of protein disulfide isomerase (PDI) and other thiol isomerases in AML-associated TF biology is unclear. THP1 cells and peripheral blood mononuclear cells (PBMCs) from healthy controls or AML patients were analyzed for thiol isomerase-dependent TF production under various experimental conditions. Total cellular and membrane TF antigen, TF PCA and TF mRNA were analyzed by ELISA, flow cytometry, clotting or Xa generation assay and qPCR, respectively. PBMCs and THP1 cells showed significant insulin reductase activity, which was inhibited by bacitracin or rutin. Co-incubation with these thiol isomerase inhibitors prevented LPS-induced TF production by CD14-positive monocytes and constitutive TF expression by THP1 cells and AML blasts. Downregulation of the TF antigen was mainly restricted to the cryptic pool of TF, efficiently preventing phosphatidylserine-dependent TF activation by daunorubicin, and at least partially regulated on the mRNA level in LPS-stimulated monocytes. Our study thus delineates a complex role of thiol isomerases in the regulation of myeloid TF PCA, with PDI being a promising therapeutic target in the management of AML-associated coagulopathies.

## 1. Introduction

Acquired coagulation disorders are a common complication in patients with acute myeloid leukemia (AML). Although particularly frequent in acute promyelocytic leukemia (APL), clotting abnormalities also occur in other AML subtypes, mainly of (myelo-) monocytic differentiation, and can manifest with both hemorrhagic and thromboembolic events [1,2]. While bleeding occurs in up to 60% of AML patients at initial presentation [3], up to 10% develop venous thromboembolism (VTE) [4,5,6,7]. Bleeding in AML patients is caused by disease- or treatment-related thrombocytopenia and complex systemic coagulopathies, such as excessive fibrinolysis, disseminated intravascular coagulation (DIC), or non-specific proteolysis [1,2]. In this regard, aberrant expression of tissue factor (TF) by transformed myeloblasts or pro-inflammatory monocytes significantly contributes to the procoagulant state [1,2,8,9].

TF is a 42-kDa transmembrane protein and the principal initiator of the coagulation protease cascade through complex formation with (activated) factor VII (FVII/FVIIa) [10]. TF is also involved in cell signaling via stimulation of protease-activated receptors (PARs), thus driving inflammation, angiogenesis and cancer progression [10,11].

Under physiological conditions, TF is mainly expressed on perivascular cells. However, TF can also be found within the vasculature on endothelial cells, leukocytes and platelets, where it is induced by pro-inflammatory stimuli, such as bacterial lipopolysaccharide (LPS), tumor necrosis factor α (TNFα) or interleukin 6 (IL-6), and oxidative stress [10,12]. In addition, TF is constitutively expressed by many cancer cells as a result of activation or inactivation of oncogenes and tumor suppressor genes, respectively [13,14]. In line with these observations, TF is frequently found on transformed myeloblasts [8,15,16,17]. However, AML is also characterized by a highly pro-inflammatory state with increased production of various inflammatory cytokines, such as IL-1β, IL-6, or TNFα [2,18]. These cytokines drive disease progression in the hematopoietic stem cell niche, but also induce TF expression in non-transformed monocytes [2,9,18].

On cell membranes, the vast majority of TF resides in a non-coagulant, functionally inactive state [19]. This cryptic TF can be rapidly converted into its procoagulant isoform by different, synergistic mechanisms. These include alterations of the membrane phospholipid composition with increased exposure of negatively charged phosphatidylserine (PS) and thiol-disulfide exchange reactions, which eventually result in oxidation of an allosteric disulfide bond within the TF extracellular domain and which are mediated by thiol isomerases, such as protein disulfide isomerase (PDI) [19,20,21].

PDI is predominantly expressed in the endoplasmic reticulum (ER) of eukaryotic cells, but it is also retained on cell surfaces or released into the vasculature upon endothelial or platelet activation [22,23,24,25]. While PDI mediates protein folding in the ER, intravascular PDI and several other thiol isomerases, such as endoplasmic reticulum protein 5 (ERp5), ERp57, ERp72, or thioredoxin, participate in thrombus formation by modulating platelet adhesion and aggregation, activation of monocyte TF and fibrin formation [24,25,26,27,28,29,30,31,32,33]. Although the precise mechanisms remain incompletely understood, inhibition of thiol isomerase reductase activity was associated with promising anti-thrombotic activity in several preclinical studies [25,26,27,28,32,34]. Moreover, daily administration of the PDI inhibitor isoquercetin significantly reduced plasma levels of D-dimer and soluble P-selectin as well as platelet-associated thrombin generation in patients with solid cancers [35]. These findings point to a therapeutic potential of thiol isomerase inhibition in the management of cancer-associated thrombosis (CAT).

Since PDI has been found in relevant concentrations in AML blasts [36], we investigated whether cellular TF procoagulant function is regulated by thiol isomerases in this hematological malignancy.

## 2. Materials and Methods

### 2.1. Materials

The following reagents were obtained from commercial sources: bacitracin (TOKU-E, Bellingham, WA, USA), quercetin-3-rutinoside (rutin), daunorubicin (DNR), lipopolysaccharide (LPS, *Escherichia coli* serotype O111:B4) (all from Sigma Aldrich, St. Louis, MO, USA), antithymocyte globulin (ATG) (Fresenius, Bad Homburg, Germany), normal human plasma (NHP) (HemosIL^®^; Instrumentation Laboratory, Kirchheim, Germany), lipidated recombinant human full-length tissue factor (Innovin^®^; Siemens Healthcare, Erlangen, Germany), plasma-derived human coagulation factor X (FX) (Merck Millipore, Darmstadt, Germany), recombinant human FVIIa (NovoSeven^®^; Novo Nordisk, Bagsvaerd, Denmark), FXa chromogenic substrate BIOPHEN^®^ CS-11(65) (Hyphen-BioMed, Neuville-sur-Oise, France), inhibitory monoclonal antibodies against human TF (no. 4509; Sekisui Diagnostics, Burlington, MA, USA) and PDI (RL-90; Novus Biologicals, Littleton, CO, USA), unconjugated mouse IgG_1_ and IgG_2a_ isotype controls (both from R&D Systems, Minneapolis, MN, USA), polyclonal rabbit IgG (Sigma Aldrich), fluorescein isothiocyanate (FITC)-conjugated monoclonal antibody against CD14 (clone RMO52; Beckman Coulter, Krefeld, Germany), TF (clone VD8; Sekisui Diagnostics) and isotype-matched IgG_1_ control (clone 679.1Mc7; Beckman Coulter), phycoerythrin (PE)-conjugated monoclonal antibodies against TF (clone HTF-1; BD Biosciences, Franklin Lakes, NJ, USA) and isotype-matched IgG_1_ control (clone 679.1Mc7; Beckman Coulter), DyLight^TM^ 488-conjugated monoclonal antibody against PDI (clone 1D3) and isotype-matched IgG_1_ control (clone MOPC-21; both from Enzo Life Sciences, Farmingdale, NY, USA) and PE-conjugated annexin V (BD Biosciences).

### 2.2. Methods

#### 2.2.1. Cell Lines and Cultures

THP1 and U937 cells were from the DSMZ, German Collection of Microorganisms and Cell Cultures GmbH (Braunschweig, Germany), and maintained in an RPMI culture medium supplemented with 10% heat-inactivated fetal calf serum (FCS) at 37 °C and 5% CO_2_.

#### 2.2.2. Collection of Blood Samples

Venous blood was drawn from an antecubital vein under no or minimal stasis into plastic tubes prefilled with 3.2% (0.109 M) sodium citrate from AML patients or healthy controls.

#### 2.2.3. Isolation and Stimulation of Peripheral Blood Mononuclear Cells (PBMCs)

PBMCs were isolated from citrate-anticoagulated whole blood by density gradient centrifugation. When indicated, monocytes were subsequently purified from PBMCs by magnetic-activated cell sorting using CD14 MicroBeads (Miltenyi Biotec GmbH, Bergisch Gladbach, Germany) on an MS column. Isolated PBMCs or CD14-positive monocytes were maintained in suspension culture in RPMI medium supplemented with 20% FCS at 2 × 10^6^/mL. In typical experiments, PBMCs or monocytes were treated with phosphate-buffered saline (PBS, control) or LPS (100 ng/mL) in the presence or absence of thiol isomerase inhibitors, rutin (100 µM) or bacitracin (5 mM), for 4 h at 37 °C. Following stimulation, cells were washed, resuspended in PBS and analyzed for their procoagulant response by various techniques. For quantitative PCR (qPCR) analysis, a shorter LPS incubation period of 2 h was chosen.

#### 2.2.4. LPS Stimulation of Whole Blood

Citrate-anticoagulated whole blood was stimulated with PBS (control) or LPS (10 µg/mL) in the presence or absence of rutin (100 µM) for 4 h at 37 °C. The LPS concentration of 10 µg/mL was based on previous findings of our group and a report analyzing the (pre-) analytical variables affecting the measurement of plasma-derived microvesicle-associated TF activity [37,38]. Following stimulation, whole blood was immediately assessed for monocyte TF antigen by flow cytometry or centrifuged twice at 2050× *g* for 10 min at room temperature (RT) to obtain platelet-poor plasma (PPP). PPP was aliquoted, snap-frozen in liquid nitrogen and stored at −80 °C until isolation of microvesicles (MVs). When indicated, PBMCs were isolated from stimulated whole blood by density gradient centrifugation as described before.

#### 2.2.5. Cell Culture Experiments

In most experiments, THP1 cells or freshly isolated myeloblasts from patients with newly diagnosed AML were used to investigate the effect of thiol isomerase inhibition under various experimental conditions. Typically, cells were suspended in fresh RPMI medium supplemented with 20% FCS at 2 × 10^6^/mL and incubated in the presence or absence of thiol isomerase inhibitors, rutin (100 µM) or bacitracin (5 mM), for 24 h at 37 °C. Cells were washed and resuspended in PBS for further analysis. When indicated, washed cells were loaded with 100 µg/mL of polyclonal rabbit IgG or ATG, or physically disrupted by repeated freeze–thawing to activate cryptic TF. In other experiments, cells were exposed for 24 h to 1 µM daunorubicin (DNR) in the presence or absence of rutin or bacitracin. Short-term incubation of cells for 30 min were carried out in PBS.

#### 2.2.6. Single-Stage Clotting Assay

Cellular procoagulant activity (PCA) was measured by single-stage clotting assay using a KC10 coagulation instrument (Amelung, Lemgo, Germany). Briefly, 100 µL of cell suspension (2 × 10^6^/mL) was mixed 1:1 (vol:vol) with NHP for 2 min at 37 °C before the addition of 50 µL of CaCl_2_ (25 mM). Times until fibrin clot formation were recorded and mean clotting times were converted into arbitrary activity units (AU) in reference to a standard curve obtained by serial dilutions (1:10–1:100,000) of Innovin^TM^. All assays were performed in duplicates, except for experiments using patient-derived myeloblasts, which were performed in triplicates. An inhibitory TF antibody (20 µg/mL) was used to demonstrate TF specificity.

#### 2.2.7. Flow Cytometry

Flow cytometry was used to measure the expression of PDI, TF and phosphatidylserine (PS) on non-permeabilized cells. Monocytes were identified in whole blood or PBMC preparations by CD14 staining. In such experiments, TF was measured by double-color flow cytometry using the function blocking HTF-1 monoclonal antibody, which specifically recognizes TF on LPS-stimulated monocytes [39,40]. In cell line experiments, single-color flow cytometry was carried out using the FITC-labeled VD8 monoclonal TF antibody. In brief, 200 µL of citrate-anticoagulated whole blood or 4 × 10^5^ cells were stained for 20 min at RT in the dark with conjugated antibodies against TF or PDI. For PS exposure, a fluorophore-labeled annexin V derivative was used according to the manufacturer’s instructions. Following incubation and lysis of erythrocytes in whole blood samples, cells were washed once with PBS and analyzed on a FACSCalibur^TM^ (BD Biosciences). Results were expressed as the proportion (%) of positive cells, which was obtained by subtracting the non-specific background measured in the presence of control IgG from the signal received in the presence of specific antibodies.

#### 2.2.8. TF ELISA

Cell lysates from THP-1 cells and PBMCs were prepared by 1% Triton X-100 protein extraction and total cellular TF antigen was measured using the Quantikine^®^ ELISA kit (R&D Systems) according to the manufacturer’s instructions.

#### 2.2.9. Thiol Isomerase Activity Assay

Cell surface thiol isomerase activity was assessed on THP1 cells, PBMCs and leukemic blasts by measuring its reductase activity using a fluorescence-based insulin reduction assay (ProteoStat PDI Assay Kit; Enzo Life Sciences).

#### 2.2.10. FXa Generation Assay

Following LPS stimulation, MVs were isolated from PPP or cell culture supernatants by double high-speed centrifugation at 16,100× *g* for 30 min at RT. Pelleted MVs were resuspended in PBS to one-third of their initial sample volume and analyzed for TF-specific activity by a two-stage chromogenic FXa generation assay, as previously described [41].

#### 2.2.11. Real-Time Quantitative PCR (qPCR)

Following stimulation of whole blood or purified PBMCs for 2 h, PBMCs were analyzed for relative changes in TF mRNA levels by real-time qPCR. To this end, stimulated cells were pelleted at 2000× *g* for 10 min at 4 °C and stored at −80 °C. The housekeeping gene GAPDH served as reference gene. Total RNA was extracted from 4 × 10^6^ PBMCs with the PureLink^TM^ RNA Mini Kit (Life Technologies^TM^, Carlsbad, CA, USA). Following reverse transcription into cDNA with the Maxima First Strand cDNA Synthesis kit for RT-qPCR (ThermoFisher Scientific, Waltham, MA, USA), quantitative PCR was performed in triplicates with the SYBR^®^ Premix Ex Taq^TM^ II Kit (TaKaRa Bio, Kusatsu, Japan) on a LightCycler^®^ 96 system (Roche, Basel, Switzerland). A total of 40 PCR cycles were carried out, each consisting of denaturation at 95 °C for 15 s, annealing at 58 °C for 5 s and extension at 72 °C for 26 s. Primer sequences for TF and GAPDH were as previously described [42]. Self-dimer formation of the primer pairs was initially excluded by gel electrophoresis (data not shown). Relative changes in TF mRNA levels were analyzed by the modified mathematical approach reported by Pfaffl [43].

#### 2.2.12. Statistical Analysis

All experiments were repeated at least three times and results are presented as mean ± standard deviation (SD). All data were normally distributed using the Shapiro–Wilk-test. When two conditions were compared, data were analyzed with the Student’s *t*-test for paired observations. For multiple comparisons, ANOVA with Tukey’s post-hoc test was carried out. Differences with a *p*-value of <0.05 were considered statistically significant.

## 3. Results

### 3.1. Thiol Isomerases Regulate Monocyte TF PCA under Pro-Inflammatory Conditions

TF-mediated coagulation abnormalities are particularly frequent in AML subtypes with (myelo-) monocytic differentiation [17]. In addition, non-transformed monocytes/macrophages participate in the generation of a hypercoagulable state. This is done through the production of TF upon stimulation by various cytokines and endotoxins released, e.g., during bacterial infections or as a result from pro-inflammatory signaling pathways within the bone marrow microenvironment [2,9]. Previous studies have shown that thiol-disulfide exchange reactions are critically involved in the regulation of TF procoagulant activity (PCA) in monocytes/macrophages [25,26,44,45]. The exact mechanisms, by which PDI and other thiol isomerases control cellular TF activity under pro-inflammatory conditions, however, remain incompletely understood. We therefore performed initial experiments to gain deeper insights into this issue.

Peripheral blood mononuclear cells (PBMCs) from healthy volunteers were stimulated with lipopolysaccharide (LPS) for 4 h to induce TF-specific PCA (Figure 1A). Freshly isolated PBMCs expressed robust insulin reductase activity, which was similar to that of purified PDI (Figure 1B). Bacitracin, a more non-specific extracellular thiol isomerase inhibitor, dose-dependently inhibited recombinant PDI (Appendix A) and also interfered with PBMC-associated insulin reductase activity (Figure 1B).

To further elucidate the role of thiol isomerases in the regulation of monocyte TF PCA under pro-inflammatory conditions, LPS stimulation of PBMCs was carried out in the presence or absence of bacitracin. While co-incubation with bacitracin had no effect on controls (Figure 1C), it reduced the PCA of LPS-stimulated PBMCs in a concentration-dependent manner (Figure 1C and Appendix A). In addition, bacitracin prevented TF antigen expression on CD14-positive monocytes (Figure 1D). Importantly, bacitracin did not amplify phosphatidylserine (PS) membrane exposure on LPS-treated monocytes (Appendix A), indicating that regulation of TF production by bacitracin did not involve cytotoxic effects.

Taken together, these findings indicate that inhibition of cell-surface thiol isomerase activity by bacitracin inhibits LPS-induced monocyte TF production.

### 3.2. Rutin Prevents LPS-Induced TF Production by PBMCs

In contrast to bacitracin, quercetin-3-rutinoside (rutin) is a more specific, reversible PDI inhibitor [28,46]. Consistently, rutin potently inhibited PDI insulin reductase activity at saturating concentrations of >50 µM (Appendix A). Similar to bacitracin, co-incubation with rutin had no effect on controls (Appendix A), but significantly inhibited LPS-induced expression of TF PCA in PBMCs (Figure 2A) and of TF antigen in monocytes (Figure 2B). The inhibitory effect of rutin was concentration- and time-dependent (Appendix A) and not explainable by increased cytotoxicity (Appendix A). Co-incubation with rutin also significantly reduced total cellular TF antigen in LPS-treated PBMCs (Figure 2C), favoring decreased TF synthesis or increased TF shedding on microvesicles (MVs) over TF internalization as the underlying mechanism. To further investigate this issue, we analyzed cellular TF mRNA and MV-associated TF PCA. Both TF mRNA (Figure 2D) and MV TF PCA (Figure 2E) were significantly reduced upon co-incubation of PBMCs with LPS and rutin. Of note, no TF antigen was detected in MV-depleted cell culture supernatants (data not shown), further arguing against increased TF shedding as the underlying mechanism.

Collectively, these data indicate that prevention of LPS-induced monocyte TF production by rutin involves decreased transcription of the *F3* gene or decreased stability of the TF mRNA.

Erythrocytes and platelets also show significant PDI surface expression [47,48]. To investigate whether other blood cells interfere with thiol isomerase-mediated regulation of monocyte TF, whole blood was stimulated with LPS in the presence or absence of rutin. Similar to previous findings, rutin significantly decreased LPS-induced release of MV TF PCA (Figure 2F). Moreover, when PBMCs were isolated from LPS-stimulated whole blood, cellular TF PCA (Figure 2G), TF antigen, as analyzed by flow cytometry (Figure 2H and Appendix A) or ELISA (Appendix A), and TF mRNA (Appendix A) were significantly reduced upon co-incubation with rutin.

Thus, inhibition of thiol isomerase reductase activity by rutin prevents monocyte TF production under pro-inflammatory conditions.

### 3.3. Rutin and Bacitracin Regulate Endogenous TF Expression in Monocytic Cells

We next asked whether thiol isomerases regulate monocyte TF production in the absence of strong pro-inflammatory stimuli, such as LPS. To this end, CD14-positive monocytes were isolated from PBMCs and maintained in suspension culture for 24 h, which resulted in the detection of cell-surface TF antigen by flow cytometry (Figure 3A,B). Consistent with our previous findings, endogenous TF expression was markedly reduced upon co-culture of monocytes with rutin or bacitracin (Figure 3A,B).

In AML, aberrant gene expression drives TF production by transformed myeloblasts [13,14]. We thus investigated whether thiol isomerases also regulated constitutive TF expression in the monocytic leukemia cell line, THP1. Similar to freshly isolated PBMCs, THP1 cells showed increased insulin reductase activity, which was inhibited by rutin (Appendix A), thus suggesting the presence of cell surface PDI activity. Constitutive TF expression by THP1 cells was confirmed by flow cytometry (Appendix A) and clotting assay (Appendix A). As previously reported [44], most of the TF on unperturbed THP1 cells resided in a functionally cryptic, inactive state. Following incubation of THP1 cells with rutin or bacitracin for 24 h, the TF antigen, as analyzed by flow cytometry (Figure 3C,D) or ELISA (Figure 3E), was significantly reduced, with bacitracin showing a more prominent effect than rutin in flow cytometry studies. Neither short-term (30 min) nor long-term (24 h) treatment with rutin affected TF PCA on intact THP1 cells (Appendix A), suggesting that TF downregulation by rutin was mainly restricted to the cryptic pool of TF. Similar to TF, the PDI antigen was downregulated from the surface of THP1 cells by overnight culture with rutin or bacitracin (Figure 3F,G). Consistent with this observation, insulin reductase activity on washed THP1 cells was significantly reduced by rutin treatment for 24 h (Figure 3H).

Taken together, these data indicate that thiol isomerases regulate TF production in monocytic cells in the absence of strong pro-inflammatory stimuli. Moreover, inhibition of thiol isomerase reductase activity is associated with concomitant downregulation of TF and PDI from the surface of THP1 cells.

### 3.4. Rutin Prevents TF Activation on THP1 Cells through Downregulation of Cryptic TF Expression

To explore the functional consequences of our previous observations on TF and PDI, THP1 cells were incubated with rutin for 24 h, washed and then treated with antithymocyte globulin (ATG), which rapidly activates cryptic TF on monocytic cells in a PDI- and complement-dependent manner [44]. Rutin significantly decreased ATG-mediated TF activation on THP1 cells (Figure 4A), a finding consistent with a reduction in the activatable cryptic pool of cell surface-expressed TF.

Cytotoxic agents activate cellular TF through induction of apoptosis/necrosis, which increases the availability of PS on the outer membrane leaflet [42,49]. THP1 cells were treated with daunorubicin (DNR) in the presence or absence of rutin or bacitracin for 24 h. As expected, DNR significantly enhanced TF-dependent PCA of THP1 cells (Figure 4B,C and Appendix A) and increased PS membrane exposure (Figure 4D). While both thiol isomerase inhibitors did not affect the PCA of controls, co-incubation with rutin or bacitracin dose-dependently prevented DNR-induced TF activation (Figure 4B,C). Importantly, antithrombotic effects of thiol isomerase inhibition could not be explained by prevention of apoptosis/necrosis, as PS exposure on rutin-treated was similar to that on buffer-treated THP1 cells (Figure 4D). To further support a role of PDI in the regulation of TF PCA under these experimental conditions, THP1 cells were treated with DNR in the presence or absence of the function-blocking PDI antibody, RL90. RL90, but not an isotype control, significantly inhibited DNR-mediated cellular TF activation (Appendix A). Inhibition of DNR-induced TF activation by bacitracin was confirmed using the monocytic leukemia cell line, U937 (Appendix A).

### 3.5. Rutin Exerts Antithrombotic Activity in Myeloblasts from AML Patients

To further explore the clinical implications of our findings obtained with monocytes and leukemic cell lines, we performed experiments using PBMCs isolated from a patient with newly diagnosed AML (patient 1 in Appendix A). While intact PBMCs showed limited TF-dependent PCA, physical disruption by repeated freeze–thawing increased cellular TF PCA 6-fold (Figure 5A), indicating the presence of significant amounts of cryptic TF on the cell surface. In addition, substantial insulin reductase activity was expressed by PBMCs, which was completely blocked by rutin or bacitracin (Figure 5B). Overnight incubation with thiol isomerase inhibitors alone had no effect on cellular PCA but prevented cellular TF activation by ATG (Figure 5C) or DNR (Figure 5D). Prevention of DNR-induced TF activation by rutin was concentration-dependent (Figure 5E) and could not be explained by inhibition of apoptosis/necrosis (Figure 5F). Similar findings were obtained in two other patients with AML (patients 2 and 3 in Appendix A). In both patients, co-incubation with rutin for 24 h alone did not affect the PCA of buffer-treated cells, but mitigated DNR-induced TF activation (Figure 6).

Collectively, these findings point to a central role of extracellular thiol isomerases, including PDI, in regulating myeloblast TF PCA in AML.

## 4. Discussion

In this study, we delineate a novel mechanism, by which thiol isomerases such as PDI regulate TF PCA in inflammatory monocytes and AML blasts. Thiol isomerase inhibition with rutin or bacitracin prevented LPS-mediated TF production by monocytes and downregulated constitutive TF expression by monocytic leukemia THP1 cells and PBMCs isolated from patients with newly diagnosed AML. Reduced de novo TF production was associated with decreased TF mRNA expression in LPS-treated monocytes and was mainly restricted to the cryptic pool of TF in THP1 cells and isolated myeloblasts, preventing its subsequent activation by ATG or DNR. Thus, PDI and other thiol isomerases might be promising new targets to prevent or treat TF-driven coagulation abnormalities in AML.

In initial experiments, we investigated the effect of thiol isomerases on TF induction in healthy monocytes (Figure 1 and Figure 2). Co-incubation of monocytes with LPS and bacitracin or rutin mitigated de novo TF production, although the inhibitory effect of rutin was less pronounced compared to that of bacitracin (Figure 1 and Figure 2). This discrepancy is likely explained by increased specificity of rutin for PDI over other vascular thiol isomerases, which also participate in thrombus formation, such as ERp5, ERp57, ERp72 or thioredoxin, while bacitracin is a more non-specific thiol isomerase inhibitor [28,31,32,33,46,50]. Of note, residual insulin reductase activity was detectable on PBMCs despite addition of 5 mM bacitracin (Figure 1B), a concentration at least 10-fold exceeding the IC_50_ of bacitracin under our experimental conditions (Appendix A), which further points to a potential contribution of other thiol isomerases in the regulation of cellular TF PCA.

Previous studies have shown that PDI and other thiol isomerases regulate cellular TF activity by posttranslational modifications of free cysteine thiols or mixed disulfides within the TF extracellular domain, a mechanism independent of de novo TF production [25,51,52,53]. Disulfide bond formation between Cys186 and Cys209 was associated with the activation of cryptic TF to its procoagulant isoform independently of PS membrane exposure [25,51,52,53,54], and TF mutants lacking these cysteine residues were significantly less potent to promote coagulation [53,55]. Alternatively, PDI might serve as an activating chaperone to enhance TF PCA; bovine liver-derived PDI enhanced Xa generation induced by both soluble and MV-associated TF independently of thiol-disulfide exchange within Cys186–Cys209, an effect that could be completely abolished by bacitracin [51]. In addition, we have previously shown that ATG can rapidly activate monocyte TF in a PDI- and complement-dependent manner and that release of TF PCA into plasma upon stimulation of whole blood with LPS is mediated by PDI [26,44]. In the latter study, co-stimulation of whole blood with LPS and a function blocking C5 antibody did not interfere with TF antigen expression on CD14-positive monocytes, but largely prevented the release of TF PCA into plasma. Since LPS-induced TF PCA release was also inhibited by rutin, these findings pointed to synergistic roles of complement and PDI in the endotoxemia model, similar to those involved in ATG-mediated monocyte TF activation.

Our current study delineates that contribution of PDI to LPS-induced production of monocyte TF PCA is more complex, since thiol isomerases are also involved in the regulation of TF protein synthesis in myeloid cells. This finding is also supported by a recent study showing that LPS-stimulated bone marrow mononuclear cells from PDI-deficient compared to wildtype mice induced significantly less TF-dependent thrombin generation [30]. Although we cannot entirely rule out the possibility that interference with thiol-disulfide exchange reactions contributed to the inhibitory effects of rutin on LPS-induced monocyte TF PCA, the overall contribution of this effect appeared to be limited, since the time-dependency analysis revealed parallel reduction of TF PCA and TF antigen under our experimental conditions (Appendix A). Moreover, reduction of the TF antigen was mainly restricted to its cryptic pool in monocytes (Figure 1C and Appendix A) and THP1 cells (Appendix A), because cellular PCA of unstimulated controls was not affected.

Although the precise mechanism by which thiol isomerases such as PDI regulate TF production remains elusive, LPS-induced PS exposure on CD14-positive monocytes and THP1 cells was not further amplified by thiol isomerase inhibitors (Figure 4D, Appendix A), which makes decreased cell viability as the underlying mechanism highly unlikely. Interestingly, several studies have reported anti-inflammatory properties for flavonoids, such as rutin, by interfering with NFκB transcriptional activity with an IC_50_ of less than 39.5 µM in vitro [56,57,58]. NFκB is implicated in the regulation of TF gene transcription [59] and critically involved in the LPS- and PAR2-mediated signaling pathways, with the latter being initiated by TF/FVIIa and TF/FVIIa/FXa complex formation [11,53,60,61]. Moreover, its transcriptional activity has been suggested to be under direct oxidoreductive control of PDI [62]. Thus, it remains tempting to speculate that thiol isomerases are involved in the regulation of TF production by an altered signal transduction, particularly by modification of NFκB transcriptional activity. However, this issue warrants further investigations.

Interestingly, MV-associated Xa generation was more efficiently reduced by rutin than cellular PCA in LPS-stimulated PBMCs, suggesting that rutin not only interfered with the supply of cryptic TF but also strongly attenuated shedding of TF-bearing MVs. Cell-injury signals such as LPS induce cellular ATP release [63], and ATP-triggered inflammatory signaling through the myelomonocytic P2X7 receptor enables shedding of procoagulant MVs into the extracellular space [64]. PDI-mediated thiol-disulfide exchange reactions within the P2X7 receptor signaling pathway are required to incorporate TF into MVs [45,64,65,66]. Accordingly, reduction of MV-associated Xa generation by rutin might additionally be explained by impaired trafficking of TF onto MVs next to reduced TF production.

PDI is upregulated in several types of cancer, including kidney, ovarian, prostate, lung, brain and germ-cell tumors, as well as lymphoma, and inhibition of its reductase activity by rutin has been associated with promising anticancer activity in vitro and in vivo [67,68]. Several mechanisms for its anti-cancer activity have been identified, including inhibition of malignant cell growth, induction of cell cycle arrest and apoptosis and modulation of angiogenesis, inflammation and oxidative stress [68]. In addition, rutin has already been demonstrated, in some earlier studies, to interfere with proliferation of murine leukemia WEHI-3 cells and of human HL-60 cells in a murine xenograft model, but the underlying mechanisms remained obscure [69,70]. In a recent study, PDI inhibition with SK053 initiated cell differentiation into mature myeloid cells in various AML cell lines and in blasts from six AML patients ex vivo [71]. Moreover, PDI has been shown to block myeloid differentiation by binding to the stem loop region of the C/EBP-α mRNA and calreticulin [36]. It is thus tempting to speculate that reduced TF production by THP1 cells might be explained by a cell differentiation process with consecutive loss of aberrant TF gene transcription. This could also explain why PS levels were similar to those of respective controls.

In our study, we analyzed PBMCs from three different AML patients, two of whom had a monocytic phenotype (Appendix A). None of these patients developed clinically relevant bleeding or thrombotic complications during treatment. In all patients, overnight incubation with rutin prevented DNR-mediated TF activation ex vivo (Figure 5D and Figure 6), suggesting a common thiol isomerase-mediated mechanism of myeloblast TF PCA regulation with potential therapeutic implications. Coagulation disorders in AML are complex. Although aberrant TF expression is not restricted to specific AML subtypes [8,15,17], we and others showed that further mechanisms next to TF expression by PBMCs and shedding of TF-bearing MVs contribute to systemic coagulation activation in AML, including acquired activated protein C resistance and release of neutrophil extracellular traps (NETs) [8,17,72,73,74]. Whether inhibition of thiol isomerases interferes with these TF-independent mechanisms has yet to be established.

A significant shortcoming of our study is that off-target effects of bacitracin and rutin other than PDI inhibition might have suppressed TF production in our experiments. The non-specific PDI inhibitor bacitracin decreases the reductase activity of PDI by competitive binding to its substrate binding b’ domain. It additionally inhibits various other enzymes required for protein folding, as well as cysteine proteases [75]. Rutin has higher specificity for PDI than bacitracin, but this flavonoid also targets other thiol isomerases, such as ERp5, ERp57, thioredoxin and thioredoxin reductase [67]. Although experiments using the inhibitory monoclonal antibody RL90 support a role of extracellular PDI in the regulation of TF production in THP1 cells (Appendix A), downregulation of cellular PDI expression or use of newer-generation PDI inhibitors with higher potency and specificity would be necessary to better differentiate between PDI and other thiol isomerases [76]. 

Our study has several additional limitations. First, we isolated PBMCs from only three AML patients, which reduces the generalizability of our findings. Second, we analyzed PBMCs instead of pure myeloblasts. Thus, we cannot rule out the possibility that obtained results were confounded by contaminating monocytes. Third, none of our patients had suffered from bleeding or thromboembolic complications, which questions the clinical relevance of our ex vivo experiments. Finally, we cannot comment on the pathophysiological implications of our findings for patients with APL, which is characterized by a particularly severe coagulopathy and in which TF production is under direct control of the PML/RAR-alpha fusion protein. Hence, further studies are needed to confirm and extend our observations in a larger patient cohort. Still, administration of isoquercetin, a rutin derivative with high oral bioavailability, was safe and showed promising antithrombotic activity in healthy subjects as well as in patients with advanced solid cancers [34,35], which makes it a promising candidate for future clinical investigations.

## 5. Conclusions

Our study delineates a novel mechanism, by which thiol isomerases regulate cellular TF PCA. By preventing aberrant TF production by AML blasts and inflammatory monocytes, inhibition of PDI and other thiol isomerases may thus be a promising therapeutic approach in the management of AML-associated coagulation disorders.

## Figures and Tables

**Figure 1 cancers-13-03941-f001:**
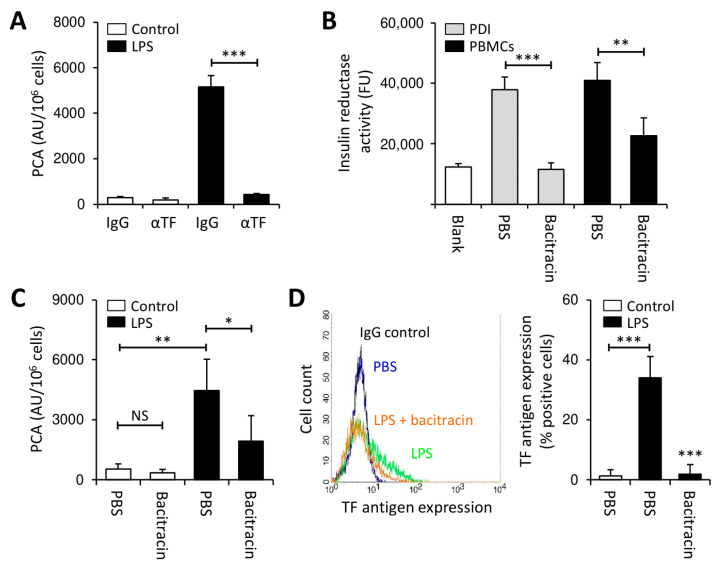
Thiol isomerases regulate lipopolysaccharide (LPS)-induced tissue factor (TF) expression in peripheral blood mononuclear cells (PBMCs). (**A**) PBMCs (2 × 10^6^/mL) were incubated with LPS (100 ng/mL) or PBS (control) for 4 h at 37 °C and subsequently analyzed for procoagulant activity (PCA) by single-stage clotting assay (mean ± SD, *n* = 6). An inhibitory TF antibody (αTF) was added in comparison to IgG isotype control to demonstrate TF specificity. (**B**) Cell-associated thiol isomerase activity was measured on freshly isolated PBMCs following incubation with PBS or bacitracin (5 mM) for 30 min using a fluorogenic insulin reductase activity assay (mean ± SD, *n* = 3). Recombinant human PDI, as provided by the manufacturer, served as positive control. (**C**,**D**) Incubation of PBMCs with LPS or PBS for 4 h at 37 °C was carried out in the presence or absence of 5 mM bacitracin. Cells were then assessed for PCA by single-stage clotting assay (mean ± SD, *n* = 6) or monocyte TF antigen expression by flow cytometry (mean ± SD, *n* = 5). *p*-values are according to Tukey’s post-hoc test (NS, not significant; *, *p* < 0.05; **, *p* < 0.01; ***, *p* < 0.001).

**Figure 2 cancers-13-03941-f002:**
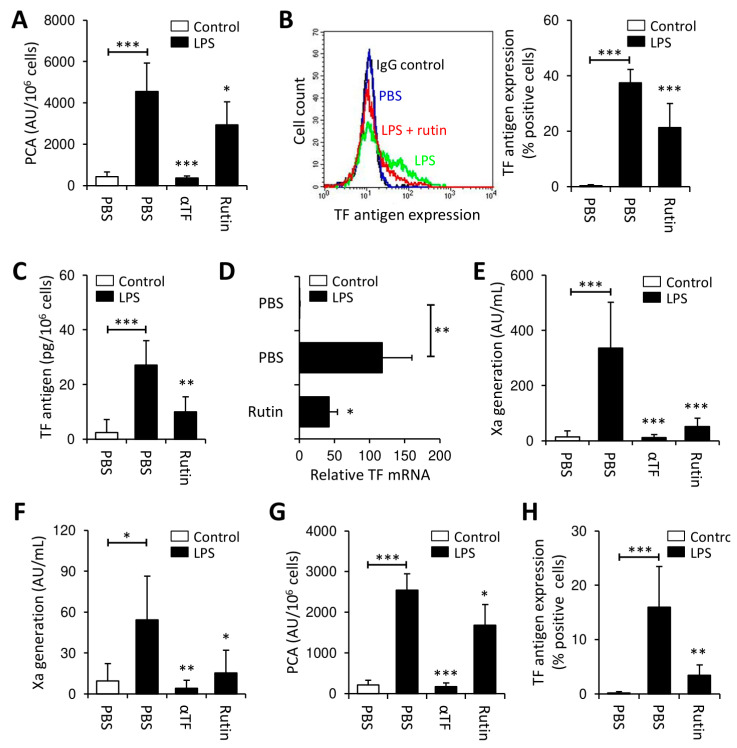
Rutin prevents LPS-induced TF production in PBMCs. (**A**–**E**) PBMCs were stimulated with LPS (100 ng/mL) in the presence or absence of the thiol isomerase inhibitor rutin (100 µM) for 2 or 4 h at 37 °C and subsequently analyzed for (**A**) TF PCA by single-stage clotting assay (mean ± SD, *n* = 13), (**B**) monocyte TF antigen by flow cytometry (mean ± SD, *n* = 10), (**C**) total cellular TF antigen by ELISA (mean ± SD, *n* = 6), (**D**) TF mRNA by quantitative RT PCR (mean ± SD, *n* = 5) and (**E**) release of microvesicle-associated TF PCA by Xa generation assay (mean ± SD, *n* = 11). (**F**,**G**) Instead of PBMCs, citrate-anticoagulated whole blood was stimulated with LPS (10 µg/mL) or PBS in the presence or absence of rutin (100 µM) for 4 h at 37 °C. (**F**) Microvesicles were isolated from platelet-poor plasma by double high-speed centrifugation and analyzed for TF PCA by chromogenic Xa generation assay (mean ± SD, *n* = 8). (**G**) PBMCs were isolated by density gradient centrifugation and analyzed for PCA by single-stage clotting assay (mean ± SD, *n* = 5). (**H**) TF antigen expression on CD14-positive monocytes was measured by flow cytometry (mean ± SD, *n* = 8). A representative histogram is shown in the Appendix A. *p*-values are according to Tukey’s post-hoc test (*, *p* < 0.05; **, *p* < 0.01; ***, *p* < 0.001).

**Figure 3 cancers-13-03941-f003:**
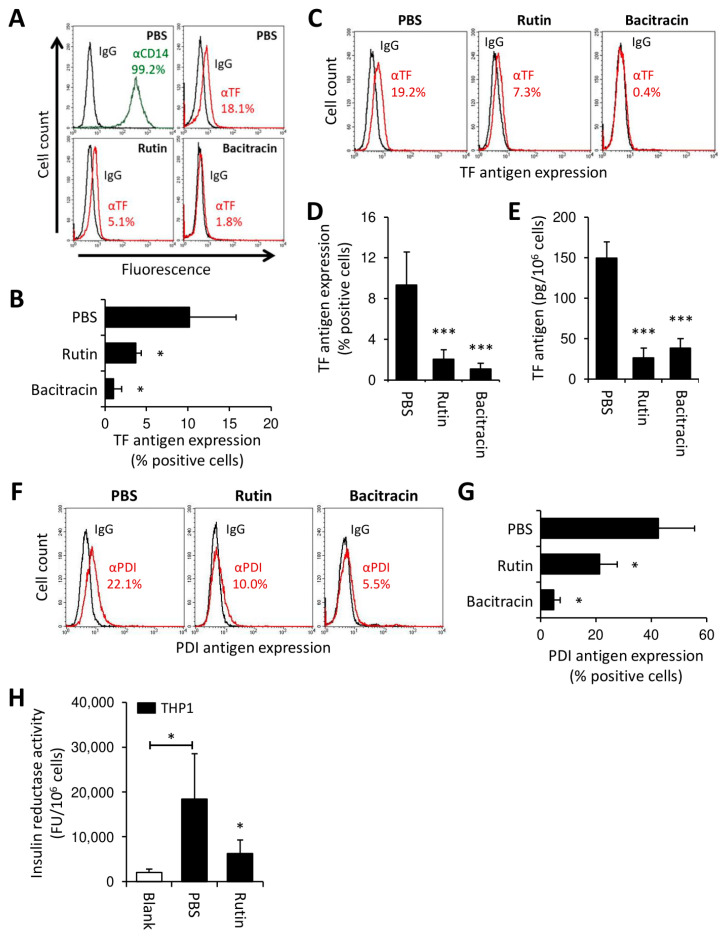
Rutin and bacitracin downregulate PDI and TF in monocytes and THP1 cells. (**A**,**B**) CD14-positive monocytes were isolated from PBMCs and maintained in suspension culture for 24 h at 37 °C in the presence of PBS (control), rutin (100 µM) or bacitracin (5 mM). Cells were subsequently analyzed for TF antigen by flow cytometry (mean ± SD, *n* = 4). (**C**–**H**) THP1 cells were incubated with PBS, rutin (100 µM) or bacitracin (5 mM) for 24 h at 37 °C, washed and analyzed for TF antigen by flow cytometry (mean ± SD, *n* = 11) (**C**,**D**), total cellular TF antigen by ELISA (mean ± SD, *n* = 3) (**E**) and for PDI antigen by flow cytometry (mean ± SD, *n* = 3) (**F**,**G**). In addition, rutin-treated THP1 cells were assessed for insulin reductase activity (mean ± SD, *n* = 6) (**H**). Representative histograms are shown in panels (**A**,**C**,**F**). Corresponding summary statistics are shown in panels (**B**,**D**) and (**G**). *p*-values are according to Tukey’s post-hoc test (*, *p* < 0.05; ***, *p* < 0.001).

**Figure 4 cancers-13-03941-f004:**
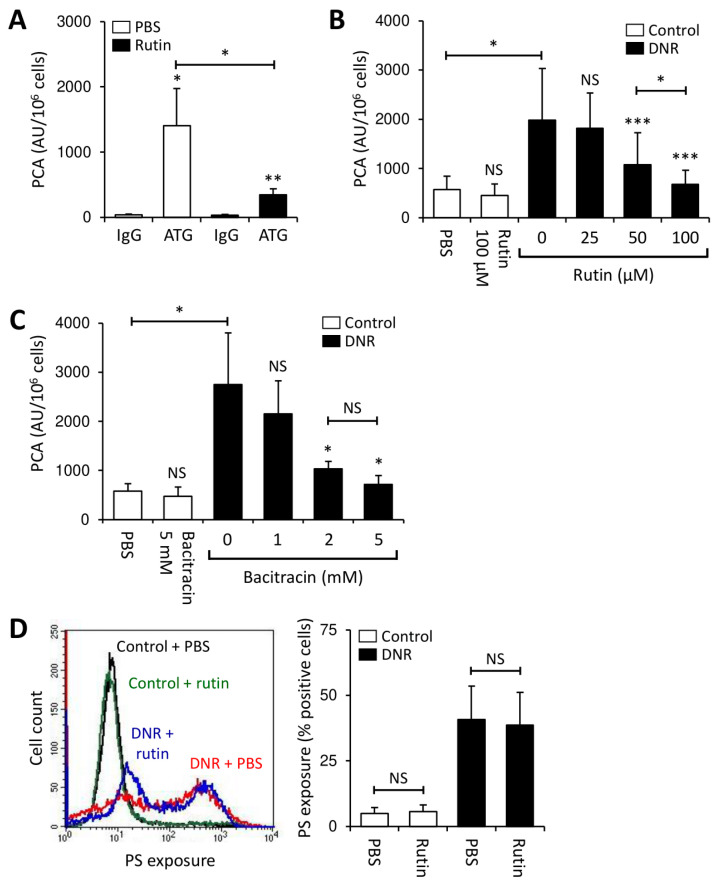
Rutin and bacitracin prevent activation of cryptic TF on THP1 cells. (**A**) THP1 cells were incubated with PBS or rutin (100 µM) for 24 h at 37 °C, washed and subsequently loaded with 100 µg/mL ATG or rabbit IgG for 15 min at room temperature. Cells were washed and mixed 1:1 (vol:vol) with normal human plasma (NHP). Following incubation for 5 min at 37 °C, cellular PCA was analyzed by single-stage clotting assay (mean ± SD, *n* = 6). (**B**,**C**) Following incubation of THP1 cells with 1 µM daunorubicin (DNR) for 24 h at 37 °C in the presence of increasing concentrations of rutin (0–100 µM) (mean ± SD, *n* = 15) (**B**) or bacitracin (0–5 mM) (mean ± SD, *n* = 6) (**C**), cells were analyzed for PCA by single-stage clotting assay. (**D**) PS exposure on THP1 cells treated with PBS (control) or 1 µM DNR in the presence or absence of 100 µM rutin was measured by flow cytometry. A representative histogram and summary statistics are shown (mean ± SD, *n* = 3). *p*-values are according to Tukey’s post-hoc test (NS, not significant; *, *p* < 0.05; **, *p* < 0.01; ***, *p* < 0.001).

**Figure 5 cancers-13-03941-f005:**
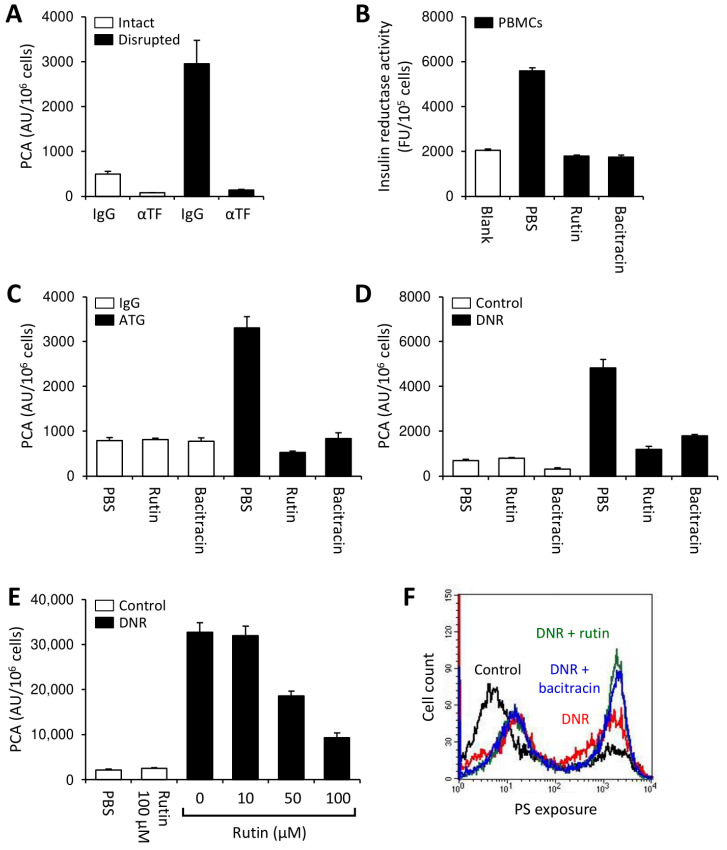
Activation of cryptic TF on PBMCs from a patient with acute myeloid leukemia (AML) is prevented by rutin and bacitracin. Whole-blood PBMCs were isolated from a patient with newly diagnosed AML and analyzed for PCA both before and after repeated freeze–thawing (**A**) and for insulin reductase activity in the presence or absence of 100 µM rutin or 5 mM bacitracin (**B**). (**C**) Following incubation with rutin (100 µM) or bacitracin (5 mM) for 24 h at 37 °C, ATG-mediated TF activation was measured as described before. (**D**) PBMCs were incubated with 1 µM DNR in the presence of 100 µM rutin or 5 mM bacitracin for 24 h at 37 °C before cell-associated PCA was measured by single-stage clotting assay. (**E**) Concentration dependency of the effect of rutin on DNR-mediated cellular TF activation. (**F**) Flow cytometric analysis of PS exposure on control and DNR-treated PBMCs co-incubated with rutin (100 µM) or bacitracin (5 µM). Results are presented as mean ± SD of triplicate measurements.

**Figure 6 cancers-13-03941-f006:**
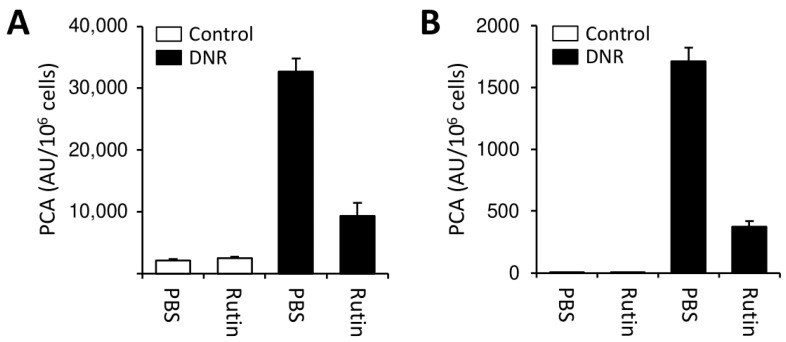
Effect of rutin on DNR-induced TF activation in PBMCs from AML patients. (**A**,**B**) Whole-blood PBMCs were isolated from two patients with newly diagnosed AML and incubated with 1 µM DNR in the presence or absence of 100 µM rutin for 24 h at 37 °C. Cells were washed and analyzed for PCA by single-stage clotting assay. Results are presented as mean ± SD of triplicate measurements.

## Data Availability

The data presented in this study are available in “Bacitracin and rutin regulate tissue factor production in inflammatory monocytes and acute myeloid leukemia blasts” or in the supplementary appendix of the same article. Original data are available on request by the corresponding author.

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
