# Peer review of "Bacitracin and Rutin Regulate Tissue Factor Production in Inflammatory Monocytes and Acute Myeloid Leukemia Blasts"

_cancers, 2021, doi:10.3390/cancers13163941_

Round 1

Reviewer 1 Report

This is a comprehensive and elegant study that delineates a novel mechanism, by which PDI regulates TF PCA in inflammatory monocytes and AML blasts. Results of this study point to  PDI as a promising new target to prevent or treat TF-driven coagulation abnormalities in AML. Therefore results from this study are highly relevant. 

Author Response

We thank the reviewer for the comments.

Reviewer 2 Report

In this study, Beckmann et al. seek to define the interplay between TF procoagulant activity (PCA) and PDI activity in AML. By taking advantage of bacitracin and rutin, two PDI inhibitors, they report that reduction of PDI reductase activity correlates with lower TF production by AML blasts and inflammatory monocytes. Based on this observation, they propose that PDI inhibition may be a promising therapeutic approach in managing AML-associated coagulation disorders.

Although the study design is elegant and the results are nicely presented and unambiguous, my main concern regards the choice of bacitracin and rutin as PDI inhibitors. Although bacitracin is very widely used as an inhibitor of PDI, it is not a specific inhibitor of PDI. Previous studies (ref. Anna-Riikka Karala and Lloyd W. Ruddock. Bacitracin is not a specific inhibitor of protein disulfide isomerase; FEBS Journal 277 (2010) 2454–2462 a 2010) have shown that bacitracin has effects on other proteins involved in protein folding and on noncatalyzed systems, with the effects on these systems being larger than the maximal effect seen on PDI. Furthermore, bacitracin can bind to and inhibit cysteine proteases. Likewise, rutin, although more specific than bacitracin, is a weak and non-specific inhibitor of PDI. It can inhibit several thiol isomerases and other enzymes located on the plasma membrane. Thus, the title: “Protein disulfide isomerase regulates tissue factor production in inflammatory monocytes and acute myeloid leukemia blasts” should be revisited based on evidence provided in the manuscript, which is: “Bacitracin and rutin regulate tissue factor production in inflammatory monocytes and acute myeloid leukemia blasts.”

In my view, to make the point that PDI inhibition is responsible for attenuation of the procoagulant phenotype seen in this study, the authors should consider ways to eliminate or downregulate PDI expression. Alternatively, they should consider using a newer generation of PDI inhibitors with higher potency and specificity (ref. Powell LE, Foster PA. Protein disulphide isomerase inhibition as a potential cancer therapeutic strategy. Cancer Med. 2021;10:2812–2825).

Author Response

We thank the reviewer very much for the valuable and thoughtful comments on our paper! We fully agree that both bacitracin and rutin have off-target effects other than PDI that might have affected our conclusions. We therefore carefully adapted our wording to encompass tissue factor (TF) regulatory effects by thiol isomerases in a more general way.

We also thank the reviewer for pointing out an important manuscript discussing the non-specificity of bacitracin and included the reference. In our discussion, we mention additional mechanisms other than PDI-mediated thiol disulfide exchange reactions, by which bacitracin and rutin may exert their effects. We also concur that the title “Bacitracin and rutin regulate tissue factor production in inflammatory monocytes and acute myeloid leukemia blasts”, as kindly suggested by the reviewer, is more precise than our initial title.

Finally, we agree with the reviewer that additional experiments would be required to delineate PDI-specific effects on TF regulation in inflammatory monocytes/AML blasts. This issue is now addressed as a significant shortcoming of our study. However, we also feel that such extensive work is outside the scope of our current manuscript, but we will certainly follow-up on this important issue.

Round 2

Reviewer 2 Report

The authors addressed most of my concerns. Also, I would like to congratulate them on their excellent work. 

To be coherent with the new title and limitations of the study, I encouraged the authors to consider two suggestions for the simple summary section. 

First: 
"...We show that thiol isomerase inhibition with bacitracin or rutin prevents..." may be changed with ..."We show that bacitracin and rutin, two pan-inhibitors of the PDI family, prevent..."

Second:
"....of myeloid TF PCA, with PDI being a promising therapeutic target..." may be changed with "...myeloid TF PCA, with the most abundant PDI being a promising therapeutic target..."

Author Response

We thank the reviewer for the encouraging comments and have changed the simple summary accordingly. Some additional minor changes have been made to the simple summary to adhere to the 150 word count limit.